# Identifying Fall Risk Predictors by Monitoring Daily Activities at Home Using a Depth Sensor Coupled to Machine Learning Algorithms

**DOI:** 10.3390/s21061957

**Published:** 2021-03-11

**Authors:** Amandine Dubois, Titus Bihl, Jean-Pierre Bresciani

**Affiliations:** 12LPN-CEMA Group (Cognition-EMotion-Action), Lorraine University, EA 7489, F-57070 Metz, France; 2Department of Neuroscience & Movement Science, University of Fribourg, CH-1700 Fribourg, Switzerland; jean-pierre.bresciani@unifr.ch; 3Head of “Frailty Care”, Cantonal Hospital, CH-1700 Fribourg, Switzerland; Titus.Bihl@h-fr.ch; 4Laboratoire de Psychologie et NeuroCognition, CNRS, Grenoble Alpes University, LPNC UMR 5105, F-38000 Grenoble, France

**Keywords:** fall prevention, monitoring at home, elderly people, machine learning algorithms, depth camera

## Abstract

Because of population ageing, fall prevention represents a human, economic, and social issue. Currently, fall-risk is assessed infrequently, and usually only after the first fall occurrence. Home monitoring could improve fall prevention. Our aim was to monitor daily activities at home in order to identify the behavioral parameters that best discriminate high fall risk from low fall risk individuals. Microsoft Kinect sensors were placed in the room of 30 patients temporarily residing in a rehabilitation center. The sensors captured the patients’ movements while they were going about their daily activities. Different behavioral parameters, such as speed to sit down, gait speed or total sitting time were extracted and analyzed combining statistical and machine learning algorithms. Our algorithms classified the patients according to their estimated fall risk. The automatic fall risk assessment performed by the algorithms was then benchmarked against fall risk assessments performed by clinicians using the Tinetti test and the Timed Up and Go test. Step length, sit-stand transition and total sitting time were the most discriminant parameters to classify patients according to their fall risk. Coupling step length to the speed required to stand up or the total sitting time gave rise to an error-less classification of the patients, i.e., to the same classification as that of the clinicians. A monitoring system extracting step length and sit-stand transitions at home could complement the clinicians’ assessment toolkit and improve fall prevention.

## 1. Introduction

As we age, our functional abilities tend to decline. Several factors, or any combination thereof, can underlie this decline, such as sensorimotor deficits associated with the aging of our sensory systems (e.g., proprioceptive, vestibular, and/or visual systems), strength loss, or cognitive impairment, to name a few. This functional decline is notably characterized by an impaired control of balance, an altered gait pattern, reduced mobility, and falls. According to the World Health Organization, “28–35% of people aged 65 and over fall each year, increasing to 32–42% for those over 70 years of age” [1]. They also report that “Falls lead to 20–30% of mild to severe injuries, and are underlying cause of 10–15% of all emergency department visits... Subsequently to falls, 20% die within a year of the hip fracture”. Fall frequency increases with age and frailty level [2], usually from the age 65 on.

In practice, fall risk is evaluated using questionnaires and clinical tests, such as the unipedal stance test (standing on one foot for as long as possible), the Timed Up and Go test (to get up, walk, turn and sit down in less than 13.5 s), the Tinetti test (balance and mobility exercises) or the Berg test (static and dynamic balance) [3,4,5,6]. However, the current clinical assessment of fall risk has limitations. First, assessment methods and tests can vary between examiners and clinical settings [7,8,9]. In addition, the assessment can be stressful for the patient, who is observed by the clinicians [10]. Finally, patients are examined infrequently, usually after the first fall occurrence, when they are already frail. In line with this, the assessment of a person’s performance at a given time in a clinical environment may not be representative of the person’s true abilities. In that context, fall prevention would greatly benefit from a system that can continuously monitor a person in his/her ecological environment, e.g., at home, and assess the evolution of his/her fall risk.

Many studies have explored the possibility to prevent fall risk using different types of sensors (for a review: [11]). However, few systems were assessed in ecological environments. To actually implement a monitoring system assessing fall risk at home, one needs to identify the behavioral parameters that best discriminate high fall risk from low fall risk individuals. Some authors investigated the effectiveness of body-worn sensors for activity monitoring in ecological environment (see [12] for a review). Some of these authors proposed an accelerometer-based method to assess specifically the fall risk of elderly people at home. Weiss et al. [13] performed three days of measurements at home. They showed that the gait variability of fallers is significantly higher along the vertical axis and significantly lower along the mediolateral axis. Rispens et al. [14] investigated the assessment potential of trunk acceleration recordings in daily life activities. Their participants wore a triaxial accelerometer for two weeks to reliably estimate gait characteristics. They showed that local dynamic stability, interstride variability, gait speed, and gait symmetry are good parameters to discriminate fallers from no-fallers. Kaye et al. [15] installed passive infrared motion detectors on the ceiling of the monitored persons’ residence to observe possible decline of walking speed. They showed that gait speed is associated with several neuropsychological and motor deficits. Other gait parameters can be extracted using systems based on image processing. For instance, Stone et al. [16] placed a Microsoft Kinect camera in the apartment of five participants for four months. They extracted spatio-temporal parameters such as gait speed, stride length and stride duration. They monitored the evolution of these parameters over time and noticed a modification of the spatio-temporal parameters for three of the participants, one of which being a patient recovering from a surgical intervention. For a good and extensive review of the sensors that have been used and tested to predict falls in smart home environments, see the one written by Forbes et al. [17]. All above-mentioned studies mainly focused on the possibility to develop a system that can be placed at home to extract one or several behavioral parameters. The aim of our study was to identify the parameters that can be used to assess fall risk in elderly people at home, and to relate these parameters to mobility factors such as balance and gait abilities. Unlike previous studies, we did not intend to develop a home-based measurement tool. Instead, based on movements performed in daily activities, we tried to identify the behavioral characteristics and activity patterns that best discriminate high from low fall risk individuals. We focused more specifically on spatio-temporal gait parameters, because the geriatric literature shows that fall risk can adequately be assessed using these parameters [18,19,20,21]. In addition, a recent study by Piau et al. [22] showed the relationship between walking speed and fall risk. This study was based on passive infrared motion sensors fixed on the ceiling. We also included activity patterns which are less commonly measured in the literature, such as the total time spent sitting or lying. A Microsoft Kinect sensor recorded the movements of elderly people going about their daily activities in their room. Because the large majority of daily locomotor activity takes place during the day, we chose to analyse day activities. This notably allowed us to perform a more accurate estimation of the spatio-temporal parameters. Clinical tests ran prior to the recording classified the elderly according to their fall risk (high vs. low). Once labeled, the depth images provided by the sensor were analyzed combining machine learning algorithms and statistical analyses to determine which motor parameters are the most relevant to discriminate high from low fall risk individuals.

## 2. Methods

### 2.1. Participants

Thirty participants (18 females, 12 males) aged 61 to 93 (mean age 83.28 ± 8.45) and residing in a transitional care unit participated in the experiment. In this care unit, hospitalized patients follow rehabilitation programs designed to allow them to regain autonomy and return home as soon as possible. The main inclusion criteria were being aged 60 to 95 years old and being able to walk at least ten meters straight without interruption. Individuals suffering from severe dementia (as assessed by the Mini Mental State Examination score [23]) were excluded, as were those suffering from visual or cardio-pulmonary disorders. The study was approved by the local ethics committee of the University of Fribourg.

### 2.2. Data Acquisition

The patients in rehabilitation were living in their own apartment. All apartments were similar to one another, and all were single room apartments. We positioned a Microsoft Kinect sensor v2 (which is a time-of-fligth 3D sensor that produces depth images [24]) in the corner of the room, with a tilt angle of about 20°. That way, all furnitures (bed, desk, chair, armchair) as well as the different doors leading to the exit (i.e., to the corridor of the institute) and to the bathroom were in the field of view of the sensor (see Figure 1). For each patient, the monitoring lasted eight hours, from 9:30 a.m. to 5:30 p.m. Patients went about their usual daily activities, without any specific instructions.

Before placing the sensor in the participant’s room, a reference clinical assessment was made by physiotherapists using the Tinetti test [25] and the Timed Up and Go (TUG) test [26]. In the Tinetti Test, the tested person performs 17 exercises, and the clinician assigns a score ranging from 0 to 2 to the execution of each exercise. The maximum score of 28 indicates ‘no fall risk’. A score inferior to 21 corresponds to a high fall risk. In the literature, the Tinetti test has been shown to be a good indicator of fall risk in community-dwelling elderly individuals and in residents of intermediate care facilities [9,25,27]. In the TUG test, the tested person is asked to rise from a chair, to walk 3 m, to turn around, to walk back to the chair at a normal pace and to sit down. The clinical staff times the execution of the test to classify patients according to their fall risk, namely high risk vs. low/no risk. Here, participants performed the TUG test three times, and the fastest trial was used for the classification. The TUG test has been criticized by some authors [28,29]. However, this test is always widely used by healthcare professionals, and moreover, it is recommended by both the American Geriatrics Society and the British Geriatric Society [5]. The classification made by the clinicians based on the results of the Tinetti and TUG tests was used as a reference when analyzing the parameters extracted with the depth sensor in the room.

### 2.3. Preprocessing

Different algorithms automatically process depth images to identify the activity an individual is engaged in [30,31,32]. However, we wanted to obtain the ground truth regarding the current activity of the person and avoid any classification error. For that reason, depth images were manually labeled. From the labeled activities, an algorithm presented in [33] automatically extracted different parameters. The total time of ‘lying’ and ‘sitting’ activities, the total time spent out of the room, and the number of transitions (number of activity changes) were calculated. The duration of the different states (i.e., not in the room, lying and sitting) was expressed in percentage of the total duration of the monitoring. The detection of the ‘sitting’ activity was used to calculate the time and speed to sit down and to get up. When the participant was in the ‘walking’ state, the spatio-temporal parameters of gait were measured. The gait parameters were extracted from the local maxima detected on the vertical axis of the centroid of the walking person [33]. One step corresponded to the interval between two local maxima. This information allowed us to obtain the mean step length, the Coefficient of Variation (CV) of the mean step length, the mean step duration, the CV of the mean step duration, the mean pace of walking, as well as the CV of the mean pace and mean gait speed. Previous studies have shown that the algorithm used in this study measures accurately the spatio-temporal parameters compared to the GaitRite [33] and to the OptiTrack system [34].

### 2.4. Data Analysis

We used the reference clinical assessment performed by healthcare professionals to classify the patients as having a high or a low/no risk of fall. Participants with a Tinetti test score of less than 21 points were considered as having a high fall risk. A Tinetti score of 21 or less has been shown to identify fall risk with a 85% sensitivity and a 56% specificity in patients with dementia [35], and and with a 74% sensitivity and a 60% specificity with Huntington’s Disease patients [36]. For the TUG test, the fall risk of participants was estimated based on the fastest of their three trials, as healthcare professionals consider the best performance to be a good indicator of an individual’s abilities. The individuals who needed 13.5 s or more to perform the test were considered as having a high risk of fall, whereas the individuals who performed the test in less than 13.5 s were considered as having a low/no risk of fall. In our study, we used 13.5 s as a threshold because it was the threshold used by the healthcare professionals at the institute where we performed the study. Moreover, Shumway-Cook et al. [37] found that a cut-off value of 13.5 s resulted in a discrimination specificity of 100% and a discrimination sensitivity of 80%. Importantly, supplementary data analyses performed as control demonstrated that a cut-off threshold set between 13 s and 16.4 s would not modify the classification results. For all but two participants, the Tinetti and the TUG test provided congruent estimations regarding fall risk, i.e., an individual considered as having a high fall risk based on the Tinetti test was also considered as having a high fall risk based on the TUG test. For two participants, the two clinical tests resulted in incongruent outcomes. Specifically, the two participants would be considered as having no fall risk based on the Tinetti performance (score of 27) and as being at high risk of fall based on the performance on the TUG (17 s for one participant and 17.81 s for the other). These two participants were considered as having a high fall risk. For one of them, the rationale behind this choice was the fact that the participant fell three times in the two years preceding the monitoring. As for the second of them, and after discussions with the clinical staff, we decided that it is better to assume that the person is at risk when there is a doubt. Eventually, nine participants were considered as having no/a low fall risk, whereas twenty one participants were considered as having a high fall risk. Table 1 presents descriptive statistics about the low and high fall risk groups. 

We used supervised machine learning methods to classify the thirty patients of the data set in two classes (low vs. high risk of fall). We assessed whether and how well this sensor-based classification matched the classification performed by the clinicians using clinical tests. We tested the classifiers with one and two parameters. Each parameter extracted during the monitoring day was averaged to obtain one value by participant (see Section 2.3 for details about these parameters). We generated all combinations of 1 and 2 parameters. Each combination was evaluated using state of the art classifiers. We used the scikit-learn implementation [38] of Decision Tree (DT), AdaBoost (AB), Neural Net (NN), Naive Bayes (NB), Nearest Neighbors (KNN), Linear Support Vector Machines (SVM), Radial Basis Function (RBF) SVM, Random Forest (RF), Quadratic Discriminant Analysis (QDA). Table 2 presents the parameter values for each algorithm. The k-fold validation procedure was performed for each classifier to compare classification performance when applied to different subsets of features. The k-fold cross validation procedure was used by dividing the data set in 30 partitions corresponding to the 30 participants, so that each individual could never be simultaneously in the training and testing set. We used the average performance calculated on the k partitions to evaluate the classifier. It is important to mention here that the clinical classification was used as classifier output in the learning phase, but was obviously never used as input of the classifier. 

The sensor-extracted parameters were also analyzed with statistical tools, using the R statistical environment. For each parameter, the difference between the two groups, i.e., high and low/no fall risk, was evaluated using either a Wilcoxon Rank Sum test (when data distribution deviated from normality or/and when variance was not homogeneous between groups) or a Student *t*-test (when data was parametric). Significance level (i.e., alpha) was set at 0.05 and adjusted for multiple comparisons using Holm corrections.

## 3. Results

### 3.1. Machine Learning

Machine Learning results are shown in Table 3. This table shows the model having the best mean accuracy. The accuracy was defined as the number of correct predictions divided by the total number of predictions. The best possible score is 1, which corresponds to a perfect match between the sensor-based and the clinicians’ classification. When using a single parameter for the classification, none of the models classified the patients without error, i.e., in accordance with the classification performed by the clinicians. ‘Mean step length’ was the parameter giving the best classification score (94.44%). 

When two parameters were used for the classification, some parameters combinations gave rise to a 100% mean accuracy, i.e., a classification 100% identical to that of the clinicians. The models giving the best results always included step length as parameter. Speed to get-up and total sitting time proved the most relevant as second parameter. Figure 2a,b present a graphical representation of all combinations giving a 100% mean accuracy. A clear boundary separating the two fall risk classes can be observed. This suggests that these two parameters can be relied upon to obtain a robust and reliable classification regarding fall risk. In contrast, Figure 2c shows an example of combination for which no clear separation is to be seen, and for which the model built with the chosen parameters (‘Mean step length’ and ‘Total lying time’) is suboptimal. The neural net, Naive Bayes and Nearest Neighbors algorithms were the most effective algorithms to classify the participants according to their fall risk.

### 3.2. Statistical Analysis

Mean comparisons performed using Student’s *t*-tests and Wilcoxon rank-sum tests indicated significant differences between the ‘High risk of fall’ and the ‘Low risk of fall’ group only for the ‘Mean step length’ parameter (*p*-value = 0.0007). Figure 3 presents the relative difference between the low-risk and high-risk group for each parameter.

## 4. Discussion

We used an ambient sensor to monitor for a whole day the daily activities of thirty patients residing in a rehabilitation center. Our aim was to identify motor parameters that can be extracted from everyday-life activities and used to predict fall risk. After a labeling step, spatio-temporals parameters characterizing the gait pattern as well as the stand-sit and sit-stand transition were automatically extracted, along with other parameters such as the total lying time or the time spent out of the room. Machine learning algorithms classified the patients according to their estimated risk of fall, and this classification was benchmarked against the fall risk assessment performed by clinicians using the TUG and Tinetti test. Two combinations of two parameters resulted in a perfect classification. These combinations were: ‘Mean step length’ combined with ‘Speed to get up’ and ‘Mean step length’ combined with ‘Total sitting time’. Additional statistical analyses revealed that the individual parameter that best discriminated the two groups according to their fall risk was ‘Mean step length’. After correction for multiple comparisons, only one parameter differed significantly between the two groups. This explains why few ‘one-parameter models’ gave rise to a good mean accuracy when applying machine learning algorithms.

In the geriatric literature, spatio-temporal gait parameters are considered relevant to assess fall risk in clinical testing [18,19,20,21]. In home-monitoring studies [14,15,16], gait speed and step lengths are detected as the main parameters for identifying people with high fall risk. However, when recording the spatio-temporal parameters of gait at home, any modification of the furniture location/room configuration can alter the parameters. This is an obvious limit of recordings performed at home. Here we show that in addition to the spatio-temporal parameters, other parameters less commonly measured, such as the speed required to stand-up or the total time spent sitting, can reliably be used to improve fall risk assessment. Indeed, when combined with ‘Mean step length’, these parameters gave rise to an error-less classification of patients’ fall risk. Please note that in a ‘real’ apartment or house with several rooms, a lesser amount of activities could be monitored because the field of view of the sensor could not cover all rooms. In this case, placing the sensor in the living room would probably constitute the best option to monitor the spatio-temporal parameters of gait as well as the stand-sit and sit-stand transitions. An advantage of using these specific parameters is that they are easy to ‘robustly’ record at home.

Because of population ageing, fall prevention represents a human, economic and social issue. Currently, fall prevention mostly relies on clinical tests performed in healthcare institutions by expert clinicians after an accident (a stroke, a fall, etc.). In this pilot study, we monitored individuals going about their daily activities at home, and we identified the parameters that are the most relevant to assess their fall risk. Using these parameters notably allowed us to classify without error the monitored patients according to the clinical assessment of their fall risk. It should be noted that the present study does not show that the identified parameters exactly predict falls, which are multi-factorial events. We show that these parameters can be used to identify fall risk, as confirmed by clinical tests. The system used in this study is easy to set up. Residents have no constraint and may freely go about their activities. They do not have to think of wearing any sensor, as is the case with some other systems, as for instance accelerometer-based systems. Importantly, our system can also be used to monitor people with assistive devices, because our algorithms are based on macroscopic features, namely the centroid displacement. Finally, an important characteristic of active depth sensing is that it also works at night. In line with this, for a future study, it could be used to monitor nightly activities (e.g., number of night lifts, time at which the person gets up and down, etc.) and evaluate their relevance for frailty assessment. The results presented in this work could be used by physicians as additional information when evaluating fall risk. In particular, we believe that adding the stand-sit and sit-stand transitions parameters to the commonly used spatio-temporal parameters is a key element to evaluate fall risk at home. This study presents preliminary results that should be confirmed, notably by validating the relevance of the identified variables in less standardized home environments and over a longer monitoring period.

## Figures and Tables

**Figure 1 sensors-21-01957-f001:**
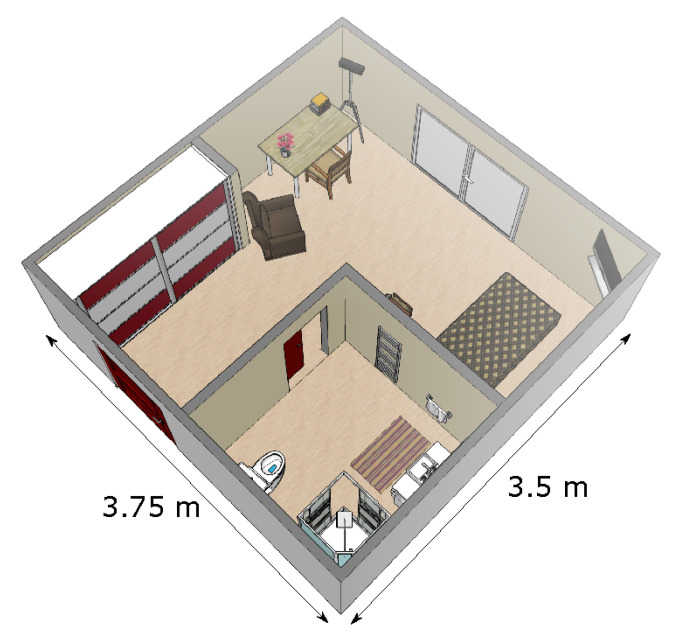
Representation of a typical patient’s room. The sensor was positioned in the corner at the top of the figure.

**Figure 2 sensors-21-01957-f002:**
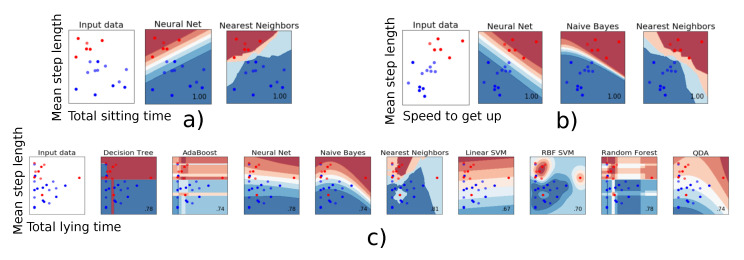
The classifier decision function between the patients who have a low (red) or a high risk of fall (blue) is represented for the combination of two parameters. (**a**,**b**): Individuals are clearly ‘split’ into two distinct categories. (**c**): These two parameters do not allow clearly classifying the individuals.

**Figure 3 sensors-21-01957-f003:**
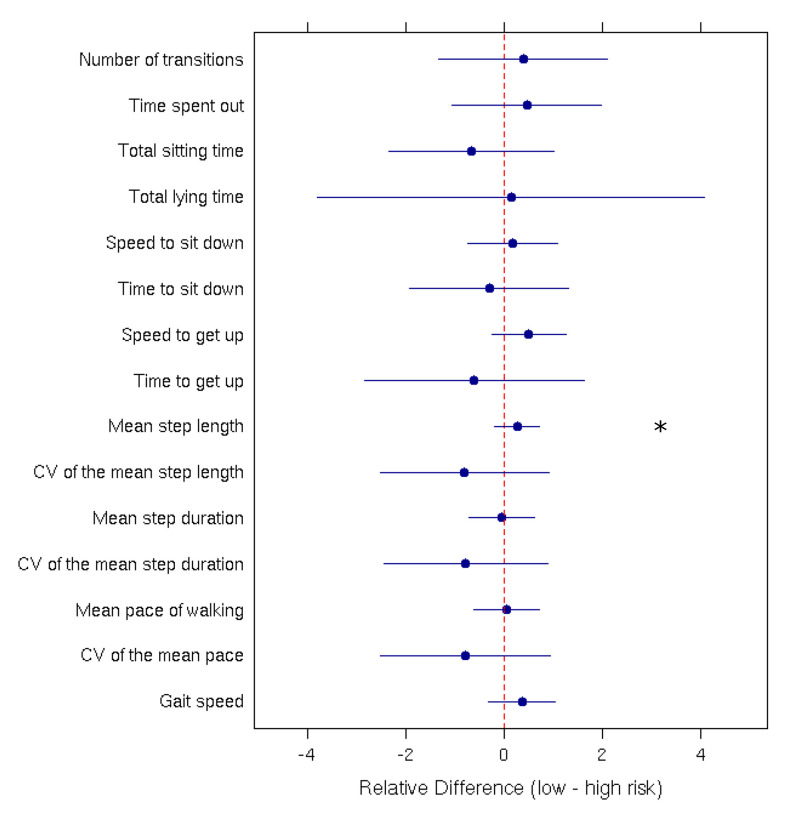
Relative difference (with the 95% confidence interval) between low and high fall-risk patients for each measured parameter. The ‘*’ indicates a significant difference between the two groups.

**Table 1 sensors-21-01957-t001:** Descriptive statistics about the low and high fall risk groups.

	Low Risk of Fall	High Risk of Fall
Number	9	21
Age/Standard deviation	79.44/6.17	85.00/8.90
Mean/Standard deviation time for TUG test (s)	11.82/1.28	23.52/5.11
Mean/Standard deviation score for Tinetti test	26.25/1.75	18.76/4.02

**Table 2 sensors-21-01957-t002:** Parameter values for algorithm Decision Tree (DT), AdaBoost (AB), Neural Net (NN), Naive Bayes (NB), Nearest Neighbors (KNN), Radial Basis Function (RBF), Random Forest (RF), Quadratic Discriminant Analysis (QDA), and Linear Support Vector Machines (SVM).

Algorithm	Parameters
DT	maximum depth = 5
AB	base estimator = DT, 50 estimators
NN	500 iterations, 100 neurons in a single hidden layer
NB	priors estimated from data
KNN	k = 3
RF	10 trees with max depth = 5
QDA	priors estimated from data
SVM	Linear kernel, C = 0.025
RBF	RBF kernel, C = 1

**Table 3 sensors-21-01957-t003:** Best parameter combinations for the two classes problem. The best classification score is presented in the fourth column. A score of 1 is the best possible score, i.e., without any classification error. The third column indicates the algorithms underlying the best classifications with Decision Tree: DT, AdaBoost: AB, Neural Net: NN, Naive Bayes: NB, Nearest Neighbors: KNN, Radial Basis Function: RBF, Random Forest: RF and Quadratic Discriminant Analysis: QDA.

Number of Parameters	Parameters	Algorithm	Mean Accuracy
1	- Mean length	DT, AB, NN, NB, KNN, RBF, RF, QDA	0.944
2	- Mean length, Total sitting time	NN, KNN	1.0
	- Mean length, Speed to get up	KNN, NN, NB	

## Data Availability

The data presented in this study are available on request from the corresponding author. The data are not publicly available due to ethical protocol restrictions.

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
