# Peer review of "Identifying Fall Risk Predictors by Monitoring Daily Activities at Home Using a Depth Sensor Coupled to Machine Learning Algorithms"

_sensors, 2021, doi:10.3390/s21061957_

Round 1
Reviewer 1 Report
In this paper, an approach to identify and predict fall risk on the basis of a combination of data acquired from depth sensors and machine learning techniques is proposed. The proposal is tested with data from Kinect sensors and 30 patients from a rehabilitation center. Some different algorithms are considered to predict fall risk and the results are promising.
The overall paper is written in a clear language, it is easy to read. In general, the significance of the paper is good since the proposed approach provide evidence of technical or theoretical novelty as well as empirical results. However, the authors should take into account the following consideration to improve the quality of the paper before publishing it:
- In the introduction, the authors should motivate their contribution by analyzing the drawbacks of the previous work in the state of the art or a related work section should be included with a good revision of the state of the art.
- A preliminaries section should be included where preliminary concepts have been presented such as some explanation on how the sensors takes data, how the algorithms work, etc. in order to help provide additional insight to the reader.
- Results section should be extended with detailed results of the different algorithms. For example, detailing the default parameter values for each algorithm, the used data for training and test the algorithms, and the validation process of each algorithm.
- Finally, the discussion and conclusions require a deeper discussion due to the proposed approach sounds very interesting and can have great potential.
Author Response
In this paper, an approach to identify and predict fall risk on the basis of a combination of data acquired from depth sensors and machine learning techniques is proposed. The proposal is tested with data from Kinect sensors and 30 patients from a rehabilitation center. Some different algorithms are considered to predict fall risk and the results are promising.
The overall paper is written in a clear language, it is easy to read. In general, the significance of the paper is good since the proposed approach provide evidence of technical or theoretical novelty as well as empirical results. However, the authors should take into account the following consideration to improve the quality of the paper before publishing it:
- In the introduction, the authors should motivate their contribution by analyzing the drawbacks of the previous work in the state of the art or a related work section should be included with a good revision of the state of the art.
We modified the introduction, notably adding references and specifying that few studies have assessed their system/algorithms in an ecological environment. In line with this, in the last paragraph of the introduction, we indicated that the referenced studies did not specifically aim at identifying fall risk indicators using an ambient sensor in ecological environment.
A preliminaries section should be included where preliminary concepts have been presented such as some explanation on how the sensors takes data, how the algorithms work, etc. in order to help provide additional insight to the reader.
We added the reference [35] in Data acquisition and a precision on the sensor (line 91-92).
Results section should be extended with detailed results of the different algorithms. For example, detailing the default parameter values for each algorithm, the used data for training and test the algorithms, and the validation process of each algorithm.
We addressed this issue by adding Table 2, which “presents the parameter values for each algorithm.”. Moreover, details about the validation procedure and constitution of training and test database are given in section “Data analysis” (we used a k-fold validation procedure).
- Finally, the discussion and conclusions require a deeper discussion due to the proposed approach sounds very interesting and can have great potential.
The discussion has been improved to highlight the strengths and potential of our approach.
Reviewer 2 Report
The paper titled “Identifying fall risk predictors by monitoring daily activities at home using a depth sensor coupled to machine learning algorithms” presents results about a study with an ambient sensor to monitor the activities of thirty patients residing in a rehabilitation center.
The authors used supervised machine learning methods to classify in two classes the thirty patients: low and high risk of fall.
Although the topic is interesting, there are several insufficiencies these need to be improved:
There is a significant lack of scientific explanations and discussions. Such discussions will significantly help the readers to the Authors’ train of thought.
Authors should provide all the details of supervised machine learning algorithms to classify in two classes the thirty patients used, for instance, topology.
Authors should compare the results with previous research in the same area in the chapters of Results and Discussion. In this way, they show better the innovation of your work.
Authors should explain how is obtained the Rate classification in your study.
The concepts and parameters important are poorly defined. There are important concepts for the understanding of the research, which are present in a general manner, without proper explanation.
Some methodological steps are mentioned only in results, which puzzle the reader even more.
Author Response
The paper titled “Identifying fall risk predictors by monitoring daily activities at home using a depth sensor coupled to machine learning algorithms” presents results about a study with an ambient sensor to monitor the activities of thirty patients residing in a rehabilitation center.The authors used supervised machine learning methods to classify in two classes the thirty patients: low and high risk of fall.Although the topic is interesting, there are several insufficiencies these need to be improved:
There is a significant lack of scientific explanations and discussions. Such discussions will significantly help the readers to the Authors’ train of thought. Authors should compare the results with previous research in the same area in the chapters of Results and Discussion. In this way, they show better the innovation of your work.
The discussion has been improved to highlight the strengths and potential of our approach.
Authors should provide all the details of supervised machine learning algorithms to classify in two classes the thirty patients used, for instance, topology.
We addressed this issue by adding Table 2, which presents the parameter values for each algorithm.
Authors should explain how is obtained the Rate classification in your study.
We replaced the term “the good classification rate” by “Mean accuracy” and we specified: “The accuracy is defined as the number of correct predictions divided by the total number of predictions”. (line 179)
The concepts and parameters important are poorly defined. There are important concepts for the understanding of the research, which are present in a general manner, without proper explanation.
Here we have a hard time identifying what the reviewer is referring to. Could the reviewer be more specific regarding the ‘concepts and parameters’ which he/she considers as ‘poorly defined’?
Some methodological steps are mentioned only in results, which puzzle the reader even more.
Here again, we do not see which ‘methodological steps’ the reviewer is referring to, especially considering that only results are presented in the ‘Results’ section.
Reviewer 3 Report
This pilot study developed a fall risk assessment performed by the algorithms against fall risk assessments performed by clinicians using the Microsoft Kinect sensors on 30 patients temporarily residing in a rehabilitation center. This study addressed an important topic and is well written. Thank you for the opportunity to review this article.
Author Response
This pilot study developed a fall risk assessment performed by the algorithms against fall risk assessments performed by clinicians using the Microsoft Kinect sensors on 30 patients temporarily residing in a rehabilitation center. This study addressed an important topic and is well written. Thank you for the opportunity to review this article.
Thank you for your comment
Reviewer 4 Report
The identification of risk factors for falls is an important issue and has been the subject of numerous studies. Proposing a non-intrusive technological prevention solution to do as well as clinical tests that are done in a spaced periods and during medical consultations is very interesting.
Many papers dealed with this issue with different approaches. Here are some examples:
Forbes, G., Massie, S. & Craw, S. Fall prediction using behavioural modelling from sensor data in smart homes. Artif Intell Rev 53, 1071–1091 (2020). https://doi.org/10.1007/s10462-019-09687-7
Amini, K. Banitsas and J. Cosmas, "A comparison between heuristic and machine learning techniques in fall detection using Kinect v2," 2016 IEEE International Symposium on Medical Measurements and Applications (MeMeA), Benevento, 2016, pp. 1-6, doi: 10.1109/MeMeA.2016.7533763.
Kong, L. Meng and H. Tomiyama, "Fall detection for elderly persons using a depth camera," 2017 International Conference on Advanced Mechatronic Systems (ICAMechS), Xiamen, 2017, pp. 269-273, doi: 10.1109/ICAMechS.2017.8316483.
A very interesting recent study (Piau A, Wild K, Mattek N, Crissey R, Beattie Z, Dodge H, Kaye J. When will my patient fall? Sensor-based in-home walking speed identifies future falls in older adults. J Gerontol A Biol Sci Med Sci. doi: 10.1093/gerona/glz128/5490440) has shown a link between the walking speed variability and the fall risks, in particular through the use of passive infrared motion detectors set on the ceiling). It's important to cite this article in the related works.
Your paper brings a lot of comment:
-The paper is very short and propose a too fast analysis of the experimentation conditions and results. So, it is not in-depth and lacks precisions and details on the data collected and analyzed.
-This is a preliminary study of 30 people but on only one day, it is not enough to draw conclusions such as those presented in the paper. How much data, what are the steps and process for the classifications ? explain clearly the different parameters used and combined and explain why (spatio-temporal gait parameters?).
-Many falls occur at night, so it is surprising that the measurements were only taken during the day.
-The study must be completed by more days to validate the results and in many configurations and people profiles.
-The references are old (2015 except one of the authors in 2017). The authors need to be updated because many works have been done since.
-Figure 1 is poor in information. It needs to be completed with the dimensions of the rooms and the angle of view of the camera.
-How do authors get the total lying time if the camera can't see the bedroom?
Finally, this document could be very acceptable but, in its current form, it needs to be completed and revised accordingly.
Author Response
The identification of risk factors for falls is an important issue and has been the subject of numerous studies. Proposing a non-intrusive technological prevention solution to do as well as clinical tests that are done in a spaced periods and during medical consultations is very interesting.
Many papers dealed with this issue with different approaches. Here are some examples:
Forbes, G., Massie, S. & Craw, S. Fall prediction using behavioural modelling from sensor data in smart homes. Artif Intell Rev 53, 1071–1091 (2020). https://doi.org/10.1007/s10462-019-09687-7
Amini, K. Banitsas and J. Cosmas, "A comparison between heuristic and machine learning techniques in fall detection using Kinect v2," 2016 IEEE International Symposium on Medical Measurements and Applications (MeMeA), Benevento, 2016, pp. 1-6, doi: 10.1109/MeMeA.2016.7533763.
Kong, L. Meng and H. Tomiyama, "Fall detection for elderly persons using a depth camera," 2017 International Conference on Advanced Mechatronic Systems (ICAMechS), Xiamen, 2017, pp. 269-273, doi: 10.1109/ICAMechS.2017.8316483.
A very interesting recent study (Piau A, Wild K, Mattek N, Crissey R, Beattie Z, Dodge H, Kaye J. When will my patient fall? Sensor-based in-home walking speed identifies future falls in older adults. J Gerontol A Biol Sci Med Sci. doi: 10.1093/gerona/glz128/5490440) has shown a link between the walking speed variability and the fall risks, in particular through the use of passive infrared motion detectors set on the ceiling). It's important to cite this article in the related works.
Thank you for these references, we added them in the introduction. Specifically:
In lines 59: “For a good and extensive review of the sensors that have been used and tested to predict falls in smart home environments, see the one written by Forbes et al. [38].”
In lines 69, “Also, a recent study by Piau et al. [37] has evidenced the relationship between walking speed and fall risk. This study was based on passive infrared motion sensors fixed on the ceiling”
In line 40, we added a review which explore the possibility to prevent the fall risk with different type of sensors. This review includes reference 3.
Your paper brings a lot of comment:
-The paper is very short and propose a too fast analysis of the experimentation conditions and results. So, it is not in-depth and lacks precisions and details on the data collected and analyzed.
We added a table (Table 2) providing more details about data analysis in the “Data analysis” section.
-This is a preliminary study of 30 people but on only one day, it is not enough to draw conclusions such as those presented in the paper. How much data, what are the steps and process for the classifications? explain clearly the different parameters used and combined and explain why (spatio-temporal gait parameters?). The study must be completed by more days to validate the results and in many configurations and people profiles.
We added the term “pilot study” in the Discussion section in order to underline that this is a pilot study based on a one day recording of 30 people.
Regarding the choice of spatio-temporal gait parameters, we specified in the Introduction section that “Also, a recent study by Piau et al. [37] has evidenced the relationship between walking speed and fall risk. This study was based on passive infrared motion sensors fixed on the ceiling”
-Many falls occur at night, so it is surprising that the measurements were only taken during the day.
We agree with the reviewer. However, in order to get a more accurate estimation of spatio-temporal parameters, we believe that it is important to start by analyzing day activities, because the large majority of the daily locomotor activity takes place during the day. We added this sentence in the last paragraph of introduction.
-The references are old (2015 except one of the authors in 2017). The authors need to be updated because many works have been done since.
We added more recent references at lines 40, 59, 69 (one study in 2019, two reviews in 2019 and 2020)
-Figure 1 is poor in information. It needs to be completed with the dimensions of the rooms and the angle of view of the camera.
We added the dimensions of the rooms on the Figure 1 and line 92, we specified the angle of view of the sensor.
- How do authors get the total lying time if the camera can't see the bedroom?
Maybe there was a misunderstanding. In the “Data acquisition” section, line 94, we indicate that the whole furniture of the room, including the bed, were in the field of view of the sensor. We added in this paragraph that acquisitions were made in single room configurations.
Finally, this document could be very acceptable but, in its current form, it needs to be completed and revised accordingly.
Round 2
Reviewer 1 Report
My mainly concerns have been addressed. Some minor tipos should be corrected.
Author Response
We corrected the tipos.
Reviewer 4 Report
Most of the requested corrections have been taken into account, in particular the addition of publications. The remark on the analysis of comparative data with different classification techniques was only partially addressed. Indeed, it seems very important in this article to detail with a specific section the measurement conditions and the results obtained by each technique. Giving only the main parameters is not sufficient and does not help to see their influence on the performance obtained.
On the other hand, the definition of the classification of high risk and low risk is still not clear. How have the calculated results been compared to the medically defined risks? According to the authors, "The accuracy was defined as the number of correct predictions divided by the total number of predictions". How do you know if the predictions are correct? You have to wait for a fall or according to the activities. But what are then the activity indicators and thresholds used? What cross-correlations are used? all this deserves explanation.
Author Response
Most of the requested corrections have been taken into account, in particular the addition of publications. The remark on the analysis of comparative data with different classification techniques was only partially addressed. Indeed, it seems very important in this article to detail with a specific section the measurement conditions and the results obtained by each technique. Giving only the main parameters is not sufficient and does not help to see their influence on the performance obtained.
Please note that we used 9 classifiers, and that we tested all combinations of 1 and 2 features among 15 possible measurements. In total, these tests led to 1080 different outcomes. For obvious reasons, we could not provide an exhaustive review of all outcomes. Rather, we chose to focus on the best outcomes, i.e., best classifiers and best combinations of features. Note also that the main objective of the manuscript was not to compare the different classifiers, but 1. to demonstrate the feasibility of automatic fall prevention at home, and 2. to identify the most relevant features to do so.
In the 2.4 section of the revised manuscript, we specified: “The k-fold validation procedure was performed for each classifier to compare classification performance when applied to different subsets of features.”
On the other hand, the definition of the classification of high risk and low risk is still not clear. How have the calculated results been compared to the medically defined risks? According to the authors, "The accuracy was defined as the number of correct predictions divided by the total number of predictions". How do you know if the predictions are correct? You have to wait for a fall or according to the activities. But what are then the activity indicators and thresholds used? What cross-correlations are used? all this deserves explanation.
The definition of low vs high risk did result from a clinical assessment performed by healthcare professionals using the Tinetti and TUG tests. This is mentioned in section 2.4, Data analysis: “We used the reference clinical assessment performed by healthcare professionals to classify the 130 patients as having a high or a low/no risk of fall.” Regarding the term ‘cross-correlations’, we assume that the reviewer really means ‘cross-validation’? The accuracy of the classification was indeed assessed using cross-validation, which is ‘classically’ used to estimate classification accuracy when working with small-size data sets.